# Site Selective Antibody-Oligonucleotide Conjugation via Microbial Transglutaminase

**DOI:** 10.3390/molecules24183287

**Published:** 2019-09-10

**Authors:** Ian J. Huggins, Carlos A. Medina, Aaron D. Springer, Arjen van den Berg, Satish Jadhav, Xianshu Cui, Steven F. Dowdy

**Affiliations:** 1Department of Cellular and Molecular Medicine, University of California San Diego, School of Medicine, La Jolla, CA 92093, USA (I.J.H.) (C.A.M.) (S.J.) (X.C.); 2Sorrento Therapeutics, San Diego, CA 92121, USA; 3Life Technologies, Thermo Fisher Scientific, Frederick, MD 21703, USA

**Keywords:** oligonucleotide therapeutics, siRNA, antisense oligonucleotides, monoclonal antibodies, antibody-siRNA conjugate (ARC), microbial transglutaminase, copper-less click

## Abstract

Nucleic Acid Therapeutics (NATs), including siRNAs and AntiSense Oligonucleotides (ASOs), have great potential to drug the undruggable genome. Targeting siRNAs and ASOs to specific cell types of interest has driven dramatic improvement in efficacy and reduction in toxicity. Indeed, conjugation of tris-GalNAc to siRNAs and ASOs has shown clinical efficacy in targeting diseases driven by liver hepatocytes. However, targeting non-hepatic diseases with oligonucleotide therapeutics has remained problematic for several reasons, including targeting specific cell types and endosomal escape. Monoclonal antibody (mAb) targeting of siRNAs and ASOs has the potential to deliver these drugs to a variety of specific cell and tissue types. However, most conjugation strategies rely on random chemical conjugation through lysine or cysteine residues resulting in conjugate heterogeneity and a distribution of Drug:Antibody Ratios (DAR). To produce homogeneous DAR-2 conjugates with two siRNAs per mAb, we developed a novel two-step conjugation procedure involving microbial transglutaminase (MTGase) tagging of the antibody C-terminus with an azide-functionalized linker peptide that can be subsequently conjugated to dibenzylcyclooctyne (DBCO) bearing oligonucleotides through azide-alkyne cycloaddition. Antibody-siRNA (and ASO) conjugates (ARCs) produced using this strategy are soluble, chemically defined targeted oligonucleotide therapeutics that have the potential to greatly increase the number of targetable cell types.

## 1. Introduction

Nucleic Acid Therapeutics (NATs), including siRNAs and AntiSense Oligonucleotides (ASOs), are a powerful class of therapeutics that have the potential to treat diseases that are refractory or undruggable by small molecule therapeutics and monoclonal antibody (mAb) therapeutics. However, due to their chemical composition of a highly charged phosphate backbone and macromolecular weight of ~6000 to ~14,000 Daltons (Da), delivery of NATs remains problematic [1]. Furthermore, naked siRNAs display poor pharmacokinetics, poor cellular uptake and poor endosomal escape [1], all of which highly restrict NATs to selective tissue types and subclasses of oligonucleotides. One strategy to address these problems is to protect and deliver siRNAs via formulation within lipid nanoparticles (LNPs) [2]. However, this approach suffers from increased toxicity and a low or poor diffusion coefficient. More recently, chemical conjugation of siRNA and ASO oligonucleotides to targeting domains has enabled binding to and cellular uptake in a limited number of therapeutically relevant cell types. For example, oligonucleotides bearing tris-GalNAc moieties that target the asialoglycoprotein receptor (ASGPR) on liver hepatocytes have shown significant successes in multiple clinical trials [3,4,5]. These encouraging results show the path forward for the targeted delivery of NATs to other, non-hepatic tissues [6].

mAbs are a highly attractive platform for generating targeted oligonucleotide therapeutics. Antibody-Drug conjugates (ADCs) are a clinically proven versatile means to specifically target highly cytotoxic payloads to numerous cancer cell types, thereby improving the therapeutic index of promising anticancer toxic agents [7,8,9]. Early ADC conjugation approaches utilized the surface amino group of lysine residues, resulting in variable DARs [10,11]. Full size mAbs possess approximately 40 reactive lysines, resulting in the potential of ~780 regioisomers for a DAR-2, 10,000 for a DAR-3 and >90,000 for a DAR-4 [12]. A representative ADC with an average DAR between 2–3 exhibits variable individual DARs from 0 to 6 or more, each with a highly different pharmacokinetic (PK) profile, as high DARs drive rapid blood clearance likely due to hydrophobicity [13,14]. Reduction of cysteine disulfides and conjugating the toxic drug to free sulfhydryls improved ADCs by conjugating drugs distal to antigen binding sites and limiting the DAR range between 0 and 8 [15,16]. However, the advent of site selective conjugation has shown dramatic improvements in the generation of consistent ADC DARs, and, consequently, PK and pharmacodynamic (PD) properties in preclinical models [12,13,17] and clinical trials [18,19]. Site selective conjugation in the generation of ADCs can be performed by several means, either enzymatic or chemical crosslinking. We favor the enzymatic Microbial Transglutaminase (MTGase) approach [20] that has been used to generate DAR-2 specific ADCs [21]. This approach requires the genetic insertion of a glutamine conjugation handle into the mAb coding sequence [21]. MTGase, an inexpensive and highly efficient bacterial enzyme involved in cell wall production, crosslinks proteins by catalyzing the formation of isopeptide bonds between the amino group of lysine and amide group of glutamine residues [20].

Here, we describe a simple conjugation procedure for the production of DAR-2 chemically defined, antibody-siRNA conjugates (ARCs). We utilized MTGase to add a bi-functional Lysine-Azide linker to mAb through heavy chain C-terminal glutamine genetic tags. siRNAs were conjugated to mAb-Linker peptides via Copper-free strain-promoted azide-alkyne cycloaddition (SPAAC) click conjugation, then purified by size exclusion chromatography (SEC) to obtain ARCs with no MTGase or siRNA contaminants. Thus, the MTGase conjugation approach allows for a rapid and quantitative site selective conjugation of oligonucleotides (siRNAs, ASOs, etc.) to mAbs to generate ARCs.

## 2. Results

### 2.1. Generation of MTGase Engineered mAbs

To test MTGase conjugation of mAbs, we first engineered the cDNA of an anti-CD33 IgG4 clinical mAb [7] to contain the MTGase conjugation tag. Synthetic cDNAs encoding the light and heavy chain variable domains were cloned upstream of the appropriate immunoglobulin constant regions (Kappa light chain, IgG4 heavy chain) into pCDNA3.1 expression plasmids. The MTGase conjugation handle, LLQGA [21], was inserted into the C-terminus of the heavy chain, followed by a translational termination. Suspension ExpiCHO-S cells were transfected with expression plasmids containing heavy chain and light chain cDNAs, incubated overnight at 37 °C, then placed at 32 °C for 10-14 days for mAb production [22]. To purify the tagged anti-CD33 mAb, cells were pelleted and culture supernatant was filtered and passed through a Protein A chromatography column followed by SEC. This approach typically yields 5 to 20 mg of purified mAb per 35 mL of culture media.

To determine the efficiency of MTGase conjugation, the degree of off-target conjugation and whether MTGase conjugation interferes with antibody specificity, we MTGase labeled the engineered anti-CD33 mAb with a Lysine-bearing fluorescent peptide, KAYA-PEG_6_-K-Fluorescein (KF) (Figure 1a). Following a 1 hr incubation at room temperature of a 1:5 molar ratio of mAb to KF peptide with 3.15 U/mL MTGase, SDS-PAGE analysis revealed a single heavy chain species of increased molecular weight (Figure 1b), consistent with a quantitative conjugation of one peptide per heavy chain or 2 per mAb. Gel imaging of fluorescein fluorescence produced signal overlapping with the conjugated heavy chain with no non-specific labeling of the light chain (Figure 1c). Furthermore, consistent with the formation of a single heavy chain conjugate, we did not detect any high molecular weight heavy chain to light chain, heavy chain to heavy chain or inter-mAb conjugation species. To determine whether modified mAb-KF retained target specificity, we stained CD33-positive acute myeloid leukemia THP1 cells and CD33-negative acute T cell leukemia Jurkat cells and performed flow cytometry analysis. The mAb-KF conjugate stained CD33-positive THP1 cells, but not CD33-negative Jurkat cells in a dose-dependent fashion, demonstrating that MTGase-mediated KF conjugation did not alter anti-CD33 mAb specificity (Figure 1d).

### 2.2. MTGase Generation of ARCs

To conjugate siRNAs to the engineered mAb, we synthesized a bi-functionalized Lysine-Azide linker peptide, K-PEG_6_-SG-K-N_3_ (KN_3_) (Figure 2a). The KN_3_ linker peptide allows for MTGase conjugation to the engineered anti-CD33 mAb via the lysine amino group and copper-free DBCO click conjugation to siRNAs via the azide group [23]. Similar to MTGase conjugation of the KF peptide above (Figure 1), a 1:5 molar ratio of mAb to KN_3_ peptide with 3.15 U/mL MTGase for 1 hr at room temperature resulted in near complete and selective conjugation of KN_3_ peptide to the anti-CD33 mAb heavy chain (Figure 2b). To separate conjugated mAb from MTGase and free linker peptide, we ran the reaction over a SEC column (Figure 2c). SEC cleanly separated all three molecules and allowed for isolation of the mAb-linker peptide conjugate in 4 fractions (Figure 2d).

Using a 5’ DBCO TEG linked phosphoramidite [24], we next synthesized an siRNA with a DBCO group on the 5’ end of the passenger strand for copper-free click conjugation (Figure 3a). To form a double stranded siRNA, the DBCO passenger strand was duplexed 1:1.1 with a guide strand. To generate an ARC, the DBCO-siRNA was incubated overnight with the purified mAb-KN_3_ conjugate in PBS + 40 mM L-Arginine at 37 °C. SEC analysis demonstrated that soluble ARC elutes separately from excess DBCO-siRNA (Figure 3b) and are primarily present in three fractions (Figure 3c). SDS-PAGE analysis showed a near complete conversion of heavy chain-KN_3_ conjugate to heavy chain-KN_3_-siRNA conjugate (Figure 3d). Importantly, we did not detect any off-target MTGase or DBCO conjugation to the light chain (Figure 3d). In addition, the heavy chain-KN_3_-siRNA conjugate displayed an apparent molecular weight shift of approximately 15 kDa, consistent with conjugation of 1 siRNA duplex per heavy chain, indicating the site selective generation of DAR-2 ARCs.

### 2.3. ARC Binding to CD33-Positive Cells

Conjugation of molecules to mAbs can alter their binding avidity [25,26,27,28]. To ensure that our site selective siRNA conjugation approach to the C-terminus of the heavy chain did not interfere with binding, we treated CD33-positive THP1 cells and assayed for binding by FACS (Figure 4). Both anti-CD33 mAb and the purified anti-CD33 ARC avidly and essentially equally bound THP1 cells. In contrast, treatment with the control FITC labeled anti-human secondary antibody only showed a minor increase above untreated cells. These observations demonstrate the ability of site selective siRNA conjugation to generate a DAR-2 ARC that retains all of the antigen binding avidity of the parent un-conjugated mAb.

## 3. Discussion

NATs, including siRNAs and ASOs, have the potential to transform medicine for the treatment of a wide array of diseases, including cancer, pandemic influenza, and Alzheimer’s Disease. However, targeting and intracellular delivery remain the two main challenges to overcome to achieve this potential [1]. While LNP encapsulation has enabled delivery of siRNAs to the liver, due to their poor PK and diffusion coefficients, the NAT field is moving away from LNPs and toward single molecule, targeted conjugates. For liver diseases, the stand out tris-GalNAc targeting domain to deliver siRNAs and ASOs to liver hepatocytes is already showing significant benefit for patients in clinical trials [3,4,5]. While the success of GalNAc conjugates has inspired a gold rush of companies striving to develop liver-targeted therapies, it also raises the possibility of successfully targeting oligonucleotide therapies to extra-hepatic tissues. Through the exquisite specificity of mAbs, ADCs have already been shown to target chemical payloads to myriad cancer cell types. Early ADC approaches were assembled through random chemical conjugation that can impair PK and clearance [13,29]. To avoid the pitfalls of random chemical conjugation, the newest generation of ADCs have been designed to use site selective conjugation approaches that place the payload at specific locations with a fixed DAR number, resulting in consistent PK and manufacturing profiles [12]. We applied these lessons into the design principles of ARCs. First, ARCs utilize a genetically engineered MTGase conjugation handle inserted into the C-terminus of the heavy chain. This results in a near quantitative MTGase conjugation of the bi-functional KN_3_ linker peptide to C-termini of both heavy chains. MTGase is efficient, and as a product of the food industry it is easily purchased in large (kilogram) quantities for low cost and therefore highly scalable. High yield production in E. coli has also recently been reported, bringing down the cost of in-house production [30] and paving the way for GMP MTGase. Second, copper-free click conjugation of DBCO-siRNAs to mAb-KN_3_ linker peptide results in a highly efficient (>90%) conjugation. Together, these methods result in the consistent lot-to-lot generation of DAR-2 ARCs that does not interfere with mAb binding and has a predicted consistent PK profile.

A potential pitfall in any conjugation strategy is off-target conjugation resulting in product heterogeneity. There are multiple glutamine and lysine residues present on the surface of monoclonal antibodies; however, MTGase reactivity depends upon preferred glutamine substrate properties, neighboring residues and solvent accessibility. Limiting the MTGase units and time of the reaction results in on-target site selective conjugation. While we have observed higher order conjugate bands, these were driven by excessively high concentrations of MTGase with overnight incubation periods. By contrast, the optimized parameters discussed here result in the consistent generation of DAR-2 ARCs free from off-target conjugations, suggesting that there is a relatively wide window for on-target conjugation vs. off-target conjugation.

Despite the availability of relatively inexpensive alkyne phosphoramidites that can be used for copper-catalyzed cycloaddition, we selected a DBCO-bearing passenger strand for SPAAC to avoid the use of copper. Oligonucleotides must be chemically stabilized for successful systemic delivery [31,32] and the inclusion of phosphorothioate (PS) backbone linkages is critical for protecting oligonucleotides from nucleases and improving potency by several fold [33]. Divalent metal ions, such as copper, can promote desulfurization (oxidation) of PS linkages [34]. We have observed changes in oligonucleotide mass consistent with this phenomenon following treatment with copper(II) sulfate and sodium ascorbate (data not shown). To avoid loss or variation of oligonucleotide potency, we therefore selected DBCO as the alkyne for cycloaddition to synthesize ARCs. In this work we demonstrate efficient conjugation of DBCO-siRNA to azide-functionalized mAb-KN_3_ using a low molar ratio, indicating that this approach can be successfully used to produce defined ARCs without the drawbacks of copper-catalyzed click.

We have demonstrated the production of uniform ARCs with a DAR of 2, but our approach could be used to produce uniform ARCs with higher or lower DARs as well. By incorporating MTGase tags on mAb light chains, a DAR of 4 can be attained [21]. To generate ARCs with a DAR of 1, a monovalent binder format such as Fab or scFv could be used. Furthermore, full-size mAbs engineered with “knobs-into-holes” heterodimeric heavy chains [35] that incorporated an MTGase tag on only one chain, but not the other, could be used to produce homogeneous DAR-1 ARCs. The ability to produce uniform defined ARCs is beneficial in many ways. Importantly, it allows optimization of parameters that are difficult to control otherwise, especially the placement of the siRNA cargo. It has been shown that conjugation site plays an important role in ADC efficacy and pharmacokinetics in vivo [36] and therefore, it is reasonable to assume that the same may be true for ARCs. Furthermore, defined placement of cargo results in lower batch-to-batch variation and streamlines the purification process by eliminating the need to remove over- or under-conjugated material. Overall, this rapid, simple, and efficient strategy is an important step toward delivering oligonucleotide therapies to extrahepatic tissues. Future studies are focused on (1) whether or not the linker between the mAb and siRNA needs to be cleavable in the endosome prior to endosomal escape, which is the case for most ADCs [10], and (2) endosomal escape of the siRNA into the cytoplasm, which is the rate-limiting delivery step for achieving RNAi activity [1].

## 4. Materials and Methods

### 4.1. Peptide Sequences and Synthesis

KF peptide sequence: KAYA-PEG_6_-K-fluorescein; KN_3_ peptide sequence: K-PEG_6_-SG-N_3_. Peptides were synthesized at 25 µmol scale on rink-amide MBHA solid support using Fmoc solid phase chemistry on a Symphony Quartet peptide synthesizer (Rainin Instruments, Protein Technologies, Inc. Woburn, MA, USA) utilizing standard protected amino acids and coupling reagents (Novabiochem, Laufelfingen, Switzerland), as well as specialty residues 5(6)-carboxyfluorescein (Sigma-Aldrich, St. Louis, MO, USA), Fmoc-lys (IVDDE)-OH and 5-azidopentanoic acid (Chem-Impex International, Wood Dale, IL, USA). For synthesis of KF, after the N-terminal final fmoc removal and capping (acetylation) (1:1:2 acetic anhydrous, DIPEA, DMF), the column was washed sequentially with DMF, DCM, DMF. ivDde was removed from the side chain of the C-terminal lysine by incubating in 2% hydrazine in DMF 2 × 15 min, followed by washing and coupling to 5(6)-carboxyfluorescein. For synthesis of KN_3_, after the N-terminal final fmoc removal and capping (acetylation) (1:1:2 acetic anhydrous, DIPEA, DMF), the column was washed sequentially with DMF, DCM, DMF. ivDde was removed from the side chain of the C-terminal lysine by incubating in 2% hydrazine in DMF 2 × 15 min, followed by washing and coupling to 5-Azidopentanoic acid. Peptides were cleaved and deprotected using standard conditions (95.0% TFA, 2.5% water, 2.5% TIS) for 2 h. Crude peptides were precipitated with cold diethyl ether and purified by RP-HPLC on a 1200 Series HPLC (Agilent Technologies, Santa Clara, CA, USA) with a Prep C18 30 × 250 mm column (Agilent Technologies). Fractions with absorbance at 230 nm were analyzed by MALDI-TOF with α-CHCA matrix on a DE Voyager Pro (Applied Biosystems, Foster City, CA, USA). Fractions with the correct mass (KF—observed mass: 1313, calculated mass: 1313; KN_3_—observed mass: 905, calculated mass 920) were pooled, lyophilized, resuspended at 25 mM in pure water and stored at −20 °C.

### 4.2. Oligonucleotide Sequences and Synthesis

Sense oligonucleotide: 5′-DBCO-TEG-AGAAGAUGCUUCAGACAGAUT-3′; Antisense oligonucleotide: 5′-UCUGUCUGAAGCAUCUUCUUT-3′. Purines (A, G) were modified with 2′ *O*-methyl ribose and pyrimidines (C, U) were modified with 2′ F ribose. Both oligonucleotides retain deoxy-T on the 3′ end following cleavage from the CPG support. Oligonucleotides were synthesized on a MerMade 6 oligonucleotide synthesizer (Bioautomation, Irving, TX, USA) using commercially available phosphoramidites (Carbosynth LLC, San Diego, CA, USA; Glen Research, Sterling, VA, USA) according to manufacturer’s recommendation on Glen dT-Q-CPG 500 support (Glen Research) and deprotected by treatment with 10% diisopropylamine (DIA) in methanol 4 hr at room temperature. Oligonucleotides were purified by RP-HPLC on a 1200 Series HPLC (Agilent Technologies) with a Zorbax SB-C18 column (Agilent Technologies). Oligonucleotides were eluted with a linear gradient of acetonitrile and fractions with absorbance at 260 nm were analyzed by MALDI-TOF with THAP matrix on a DE Voyager Pro (Applied Biosystems) as well as by denaturing gel electrophoresis in methylene blue stained polyacrylamide-urea gels (15% acrylamide, 7M Urea). Fractions containing peaks with the correct mass (Sense strand-observed mass: 7948, calculated mass: 7958; Antisense strand—observed mass 6650, calculated mass 6659) were pooled, lyophilized, and resuspended at 500 nM in 50:50 acetonitrile:water.

### 4.3. Monoclonal Antibody Production and Purification

The variable regions of a monoclonal anti-CD33 antibody (hP67.6) were synthesized as gBlocks (IDT) and seamlessly cloned into expression plasmids in frame with sequences encoding the appropriate constant regions (homo sapiens IgG4 heavy chain constant region and homo sapiens kappa light chain constant region) using InFusion^®^ HD Reagent (Takara Bio USA, Mountain View, CA, USA). The heavy chain terminal lysine residue was replaced with LLQGA by PCR-mediated mutagenesis. All antibody coding sequences were confirmed by Sanger sequencing. Plasmids were transfected in ExpiCHO-S cells (Thermo Fisher Scientific, Waltham, MA, USA) according to the manufacturer’s recommendation for a 30 mL culture size in a 125 mL polycarbonate vent cap flask (Thermo Fisher Scientific) at a mass ratio of 4:1 for heavy chain and light chain plasmids. Cells were incubated overnight in a humidified 37 °C incubator with 8% CO_2_ on a shaking platform set to 140 RPM. The following day, cells were treated with ExpiFectamine CHO Enhancer (Thermo Fisher Scientific) and ExpiCHO Feed (Thermo Fisher Scientific) according to the manufacturer’s recommendation and transferred to a humidified 32 °C incubator with 5% CO_2_ on a shaking platform set to 140 RPM. 5 days after transfection, additional ExpiCHO Feed was added according to the manufacturer’s recommendation. Antibody was harvested 12 days after transfection. Briefly, the cell suspension was centrifuged at 200× *g* for 5 min at 4 °C in a 50 mL conical tube. The first supernatant was collected in a clean 50 mL conical tube and centrifuged again at 5000× *g* for 30 min at 4 °C. The second supernatant was filtered sequentially through 0.45 μm and 0.22 μm PVDF syringe filters. Antibody was isolated from clarified supernatant by Protein A affinity chromatography and further purified and buffer exchanged by SEC in phosphate buffered saline. Antibody concentration was determined by BCA Protein Assay (Thermo Fisher Scientific).

### 4.4. MTGase Conjugation

10× MTGase enzyme buffer was prepared (Sodium Chloride 500 mM, Sodium Acetate 500 mM, pH 5.8) and a working aliquot was diluted 10-fold to 1× in water. Activa MTGase enzyme (Ajinomoto North America, Inc., Itasca, IL, USA) was reconstituted in 1× MTGase enzyme buffer to achieve a working concentration of 63 mg/mL (12.6 Units/mL). 10× MTGase reaction buffer was prepared (NaCl 1.5 M, Tris-HCl pH 8.0 250 mM). The reaction was set up as follows in a 1.5 mL microcentrifuge tube: 10× MTGase reaction buffer equivalent to 1/10th the final reaction volume was added. MilliQ water was used to adjust the final volume. MTG-tagged antibody was added to a final concentration of 6.5 µM. Lys-bearing peptide was added to a final concentration of 65 µM (5:1 molar excess relative to the 2 conjugatable MTG sites on each antibody). MTGase was added to a final concentration of 3.15 U/mL. The reaction was mixed well by pipetting and incubated on a rotisserie at room temperature for one hour. Conjugation efficiency was analyzed by electrophoresis on a 10% SDS-PAGE gel (150 V, 70 min), followed by staining with Blazin’ Bright™ Luminescent UV Protein Gel Stain (Gold Biotechnology, St. Louis, MO, USA) and imaging on a UV transilluminator. Fluorescein was imaged before staining using an epifluorescence imager with an Alexa 488 filter.

### 4.5. Azide-Alkyne Click Conjugation

DBCO-bearing sense strand oligonucleotide was mixed and duplexed with 1.1 equivalents of antisense strand (to eliminate un-duplexed sense strand) by heating to 65 °C for 3 min, cooling to 22 °C, and holding for 3 min, followed by cooling to 4 °C. The amount of duplex for a ratio of 2.5:1 relative to mAb-KN_3_ peptide (2 per mAb) was dried down by vacuum evaporation and resuspended to a final concentration of 333 µM with mAb (67 µM) in phosphate buffered saline and l-Arginine (40 mM). The reaction was incubated at 37 °C for 16 h. Conjugation efficiency was analyzed by electrophoresis on a 10% SDS-PAGE gel (150 V, 70 min), followed by staining with Blazin’ Bright™ Luminescent UV Protein Gel Stain (Gold Biotechnology) and imaging on a UV transilluminator.

### 4.6. FPLC Purification of Conjugates

Crude conjugation reactions were purified by SEC using a BioLogic FPLC (Bio-Rad, Hercules, CA, USA) and a SEC650 10 × 300 gel filtration column (Bio-Rad) equilibrated in PBS with a flow rate of 1.8 mL/min. Absorbance at 230, 260 and 280 nm was measured and 0.5 mL fractions were collected across all significant peaks. The highest purity fractions were pooled and concentrated using Amicon spin filters, 30 kDa MWCO (Millipore, Burlington, MA, USA) and quantified by BCA protein assay (Thermo Fisher Scientific).

### 4.7. Cell Culture

THP1 and Jurkat cells were grown in RPMI (Thermo Fisher Scientific) supplemented with 10% Fetal Bovine Serum (Sigma-Aldrich) and Penicillin/Streptomycin (Thermo Fisher Scientific) in a humidified 37 °C tissue culture incubator with 5% CO_2_ atmosphere. Cells were split when culture density reached approximately 8.0 × 10^5^/mL by centrifuging a portion of the culture at 200× *g* for 3 min, aspirating the spent medium and resuspending the cells to a final density of 2.0 × 10^5^/mL. ExpiCHO-S cells (Thermo Fisher Scientific) were cultured in ExpiCHO Expression Medium (Thermo Fisher Scientific) in a humidified 37 °C tissue culture incubator with 8% CO_2_ atmosphere. 30 mL cultures were grown in 125 mL vented shaker flasks (Thermo Fisher Scientific) on an orbital shaker set to 140 rpm. Cells were split when density reached approximately 5.0 × 10^6^/mL by taking an aliquot of cells and diluting it to between 1.0 × 10^5^–3.0 × 10^5^ cells/mL with fresh pre-warmed culture medium.

### 4.8. Cell Staining and Flow Cytometry

For each condition tested, 1.0 × 10^5^ cells were pelleted in a microcentrifuge tube at 200 × *g* for 3 min and the supernatant was removed by pipetting. All subsequent steps were carried out on ice with ice-cold solutions. Pellets were resuspended in 100 μL of FACS buffer (PBS, BSA 1 mg/mL, EDTA 5 mM) and pelleted again. The supernatant was removed, cells were then resuspended in mAb-KF at the indicated concentrations, and incubated on ice under foil for 30 min with gentle mixing by flicking the tube every 10 min. Cells were pelleted again and stain was removed by pipetting. Cells were washed 3× by resuspending in 100 µL FACS buffer, pelleting, and aspirating. Cells were resuspended in 350 µL FACS buffer and strained through nylon mesh into clean FACS tubes on ice. Cells were assayed for fluorescein signal by flow cytometry on an LSR II (BD Biosciences, Franklin Lakes, NJ, USA) and histograms were plotted using FlowingSoftware (http://flowingsoftware.btk.fi/) and FlowJo^TM^ (FlowJo LLC, Ashland, OR, USA). For ARC binding, mAb and ARC were diluted to 25 nM in FACS buffer. THP1 cells were stained with mAb or ARC as described above and washed as described above. Goat anti-Human-FITC secondary antibody (Thermo Fisher Scientific) was diluted 1:200 in FACS buffer and applied to cells in the same manner followed by washes as described above. Staining was assayed for FITC signal by flow cytometry, as described above.

## Figures and Tables

**Figure 1 molecules-24-03287-f001:**
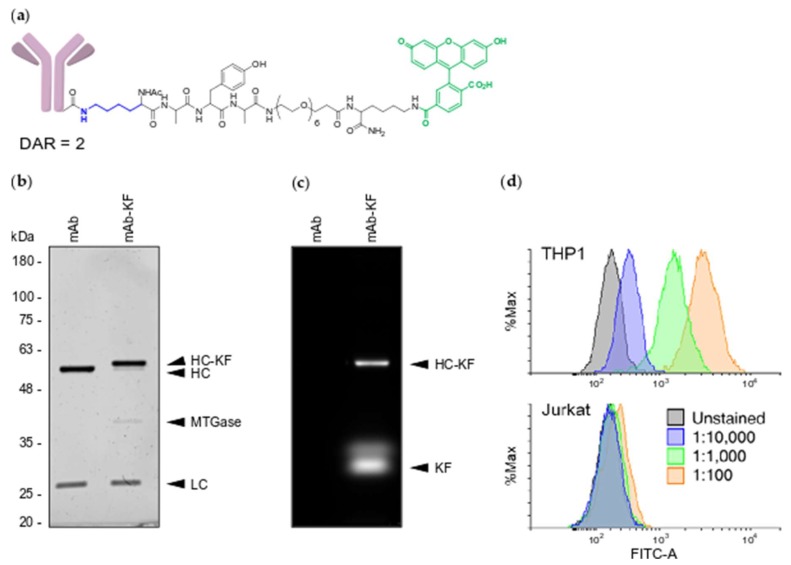
MTGase site specifically labels engineered anti-CD33 mAb. (**a**) Structure of mAb-KF peptide conjugate. Conjugated lysine is highlighted in blue and fluorescein is highlighted in green; (**b**) UV protein stained SDS-PAGE gel of unconjugated mAb and mAb-KF. Molecular weight markers are in kilodaltons (kDa). HC—heavy chain, LC—light chain, HC-KF—heavy-chain-KF conjugate, MTGase—microbial transglutaminase; (**c**) FITC epifluorescence image of the same SDS-PAGE gel. KF—free KF peptide; (**d**) Flow cytometry of mAb-KF stained THP1 (CD33-positive) and Jurkat (CD33-negative) cells. Following spin column cleanup, mAb-KF conjugate was directly diluted in stain buffer at the indicated ratios: blue—1:10,000, green—1:1,000, orange—1:100. Unstained cells are represented in gray. All plots are represented as percent maximum cell count.

**Figure 2 molecules-24-03287-f002:**
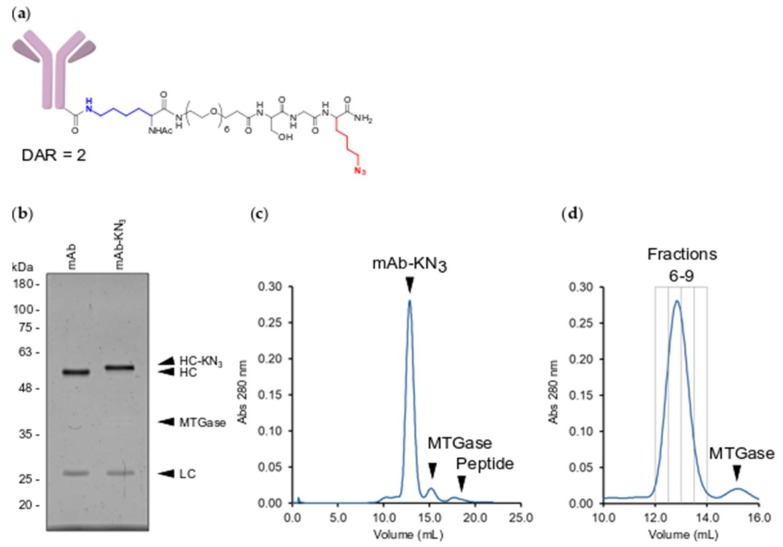
Conjugation and purification of mAb-KN_3_ conjugate. (**a**) Structure of mAb-KN_3_ peptide conjugate. Conjugated lysine is highlighted in blue and conjugatable azidolysine is highlighted in red; (**b**) UV protein stained SDS-PAGE gel of unconjugated mAb and mAb-KN_3_. Molecular weight markers are in kilodaltons. HC—heavy chain, LC—light chain, HC-KN_3_—heavy-chain-KN_3_ conjugate, MTGase—microbial transglutaminase; (**c**) SEC analysis of the crude mAb-KN_3_ conjugation reaction. mAb-KN_3_ – soluble mAb-KN_3_ conjugate, MTGase—microbial transglutaminase enzyme, Peptide—unconjugated KN_3_ peptide. (**d**) SEC purification of mAb-KN_3_ conjugate. Gray boxes indicate the elution volumes of collected fractions (6–9).

**Figure 3 molecules-24-03287-f003:**
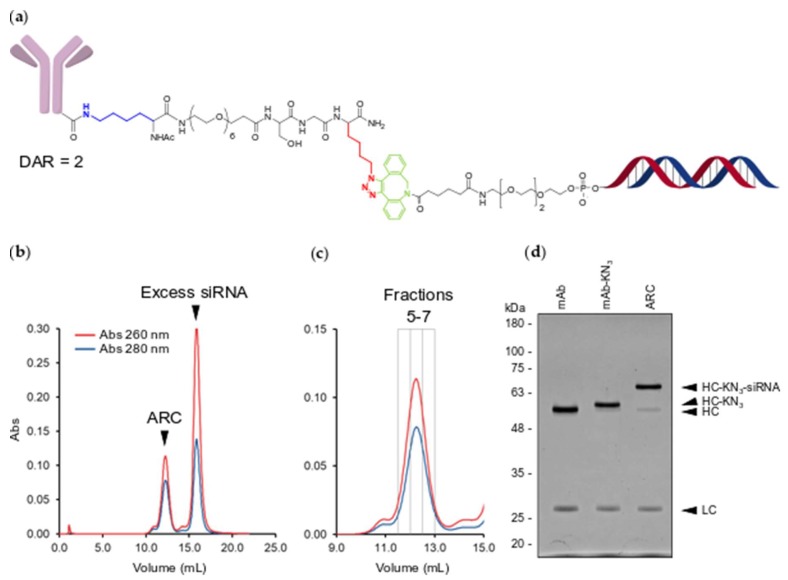
Conjugation and purification of ARC. (**a**) Structure of mAb-KN_3_-DBCO-TEG-siRNA conjugate. Conjugated lysine is highlighted in blue. Conjugated triazole (azidolysine origin) is highlighted in red. Conjugated DBCO moiety is highlighted in green; (**b**) SEC analysis of the crude ARC conjugation reaction. UV absorbance at 260 nm is traced in red and absorbance at 280 nm is traced in blue; (**c**) SEC purification of ARC. Gray boxes indicate the elution volumes of collected fractions (5–7); (**d**) UV protein stained SDS-PAGE gel of unconjugated mAb, mAb-KN_3_ and ARC. Molecular weight markers are in kilodaltons. HC—heavy chain, LC—light chain, HC-KN_3_—heavy-chain-KN_3_ conjugate, HC-KN_3_-siRNA–heavy-chain-KN_3_-siRNA conjugate.

**Figure 4 molecules-24-03287-f004:**
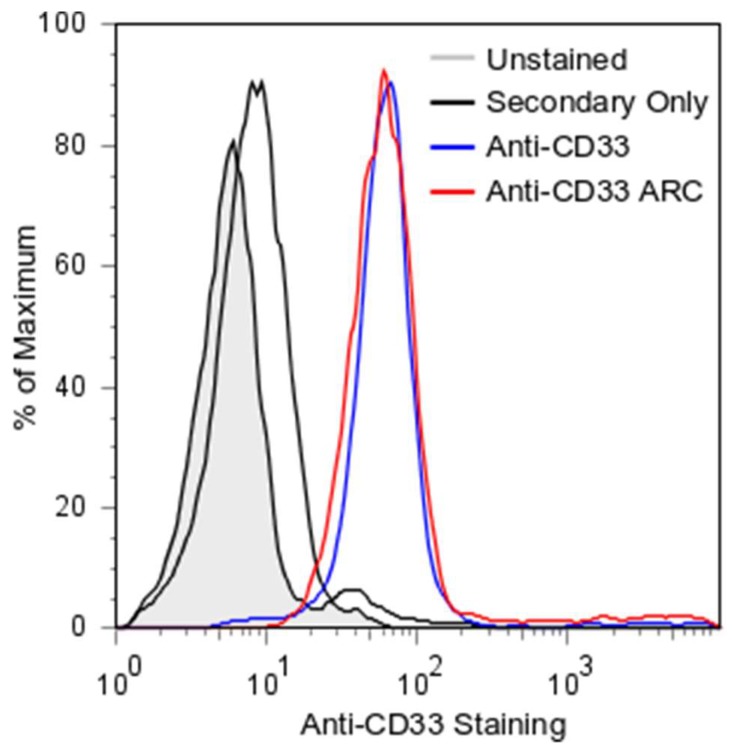
ARC Binding to THP1 Cells. Flow cytometry of mAb and ARC stained THP1 (CD33-positive) cells. Following SEC purification, mAb/ARC were applied to cells at 25 nM in FACS buffer. Secondary FITC labeled antibody was diluted 1:200 in FACS buffer. Unstained cells are represented in gray. Secondary FITC labeled antibody only is a solid black line. Anti-CD33 mAb is represented by a blue line. Anti-CD33 ARC is represented by a red line. All plots are represented as percent maximum cell count.

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
