# Peer review of "Site Selective Antibody-Oligonucleotide Conjugation via Microbial Transglutaminase"

_molecules, 2019, doi:10.3390/molecules24183287_

Round 1

Reviewer 1 Report

There is immense interest to produce antibody-siRNA/ASO conjugates.  This paper achieves this; the results are clearly presented and the authors give ample literature background/references for those news (or already) in this field.  The PI is well known and quite respected in the area of siRNA delivery, and the results are timely for those pursuing siRNA/ASO therapeutics.  Publication is recommended.    There are some points that the authors should considered.

1) the authors provide reasoning for choosing the enzymatic method described, the Cu free approach to conjugation, and the advantage of site-specific conjugation over other methods.   It would be good to comment on the scale that could in principle be achieved by the method. I feel this is important when moving to the synthesis of multi-grams of material for clinical trials.   What are the current limitations of this approach and what can be done in the future to address them?   I think it will add significantly to the manuscript if the authors were to address this important point.  

2) Experimental.   The authors cleave the oligomers using "DIA" in methanol?  what is DIA?  is this a typo?  

3) Citations are needed after these statements:

"Early ADC conjugation approaches utilized the surface 52 amino group of lysine residues that resulted in DARs from 0 to 8 or more, each with a highly different 53 pharmacokinetic (PK) profile (high DARs results in rapid clearing from the blood). Reduction of 54 cysteine disulfides and conjugating the toxic drug to the free sulfhydryl improved ADC and 55 narrowed the DAR range between 0 and 8."

4) Abstract refers to ASO-antibody conjugates.  However, ASO were not part of this study and hence it should only refer to siRNAs.  

5) despite the efforts made by the authors to define abbreviations in the text, i feel a list of abbreviations at the end of the manuscript would be quite useful when reading this manuscript. e.g., point 2 above.

Author Response

Response to Reviewer #1:

We appreciate the reviewer's positive comments and agree that "There is immense interest to produce antibody-siRNA/ASO conjugates."

#1). It would be good to comment on the scale that could in principle be achieved by the method. I feel this is important when moving to the synthesis of multi-grams of material for clinical trials.   What are the current limitations of this approach and what can be done in the future to address them?   I think it will add significantly to the manuscript if the authors were to address this important point.  

The production of engineered, site selective conjugation mAbs for clinical scale is already being performed by Seattle Genetics, Genetech and others for ADCs.  For oligonucleotide conjugation, MTGase is a food industry enzyme and as such, you can readily buy 100 kg of quantities of MTGase.  We literally bought 1 Kg of MTGase on Amazon for $100.

#2) Experimental.   The authors cleave the oligomers using "DIA" in methanol?  what is DIA?  is this a typo?

DIA = Diisopropylamine, which we neglected to define and use for ultra-mild deprotection.  Any other deprotection agent could also be used.  Thank you for pointing this out.

#3) Citations are needed after these statements:

"Early ADC conjugation approaches utilized the surface amino group...  from the blood)." "Reduction of cysteine disulfides...  DAR range between 0 and 8."

We have added references for both of these sentences.

#4) Abstract refers to ASO-antibody conjugates.  However, ASO were not part of this study and hence it should only refer to siRNAs.

We included ASOs in the abstract because the same methods are used to make ASO ARCs.  However, we have edited the text to make sure that it reads correctly for our work presented here that is focused only on siRNA ARCs.

#5) despite the efforts made by the authors to define abbreviations in the text, i feel a list of abbreviations at the end of the manuscript would be quite useful when reading this manuscript. e.g., point 2 above.

We agree and have added a short abbreviation list at the beginning of the Methods section.

Reviewer 2 Report

The manuscript describes the use of copper free click chemistry to create an antibody-siRNA conjugate. Per the authors, this method produces a homogeneous conjugate which is superior in terms of mAb binding and PK. The concept is interesting and may have a potential use.

I do have a few comments and questions:

A general scheme of the conjugation reaction is advised. At the moment each part is shown in a different figure. I am missing a proof of concept. The authors did flow cytometry to show that the Ab binds at an early stage, but it was not done for the final conjugate with the siRNA attached.

And, since the method suppose to create a superior conjugate, how is the binding activity of the final conjugate Ab-siRNA compared to the one synthesized in the common approach (ELISA or any other comparative assay to show that your homogeneous product is indeed superior?)

what about the siRNA activity? Can an in vitro assay be done to show that there is silencing?

Can the authors also discuss or mention the mechanism of cell penetration and endosomal escape? Since the Ab is large compared to the siRNA (unlike the tris-GalNAc conjugates mentioned by the authors, dx.doi.org/10.1021/ja505986a | J. Am. Chem. Soc. 2014, 136, 16958−16961) does the bond needs to be cleaved in the cytoplasm in order to have an effective silencing? (for example, Arrowhead Pharmaceuticals, employs GalNAc as a targeting and masking moiety linked to the side chain by carboxy dimethylmaleic anhydride (CDM) bond. CDM bond is a pH-sensitive linkage that is degraded quickly in an acidic environment - Molecular Therapy: Nucleic Acids Vol. 6, 2017, p. 116-132).

Author Response

Response to Reviewer #2 (our numbering):

#1) A general scheme of the conjugation reaction is advised. At the moment each part is shown in a different figure. I am missing a proof of concept.

We agree with the reviewer and presented the project outline in the graphical abstract.  Pending the editor's recommendation, we can add this into the early text.

#2) The authors did flow cytometry to show that the Ab binds at an early stage, but it was not done for the final conjugate with the siRNA attached.

We agree with the reviewer and have added a new Figure 4 and Section 2.3 to the manuscript.

#3) And, since the method suppose to create a superior conjugate, how is the binding activity of the final conjugate Ab-siRNA compared to the one synthesized in the common approach (ELISA or any other comparative assay to show that your homogeneous product is indeed superior?)

Because of the significant complications with making ARCs of DAR 0-8, and we felt that they would have no place in moving ARCs forward towards the clinic, we did not spend time generating those control constructs.  However, the problem with DAR 0-8 ADCs vs. fixed DAR ADCs in the clinics and preclinical models is well documented (see refs #10-12).

#4) What about the siRNA activity? Can an in vitro assay be done to show that there is silencing?  Can the authors also discuss or mention the mechanism of cell penetration and endosomal escape?

This study was written to present the methods to generate and purify ARCs, not to perform the in-depth investigation into their in vitro and in vivo activities, which we intend to eventually publish in a separate complete study.  Endosomal escape remains the rate-limiting step for delivery, and we are working on the synthesis of a (hopefully) universal endosomal escape domain for a future publication.

#5) Since the Ab is large compared to the siRNA (unlike the tris-GalNAc conjugates...) does the bond needs to be cleaved in the cytoplasm in order to have an effective silencing? (for example, Arrowhead Pharmaceuticals, employs GalNAc as a targeting and masking moiety linked to the side chain by carboxy dimethylmaleic anhydride (CDM) bond. CDM bond is a pH-sensitive linkage that is degraded quickly in an acidic environment.

The reviewer is correct, just like what happens with GalNAc delivery, the mAb (or GalNAc) targeting domain will need to separate from the RNAi trigger for it to transit across the endosomal membrane.  We are currently investigating and comparing linker chemistries; however, these studies will require in-depth synthesis and analyses.  Therefore, they are beyond the scope of this manuscript.

Round 2

Reviewer 2 Report

The authors made an effort to answer most of the questions. 

Per the authors, some of the issues raised were beyond the scope of this paper (siRNA activity, endosomal escape/bond in the cytoplasm).

Although I may generally accept such an argument,  I wish that the author's will make it clear also in the manuscript:  siRNA activity studies are ongoing and that endosomal escape mechanism or weather the bond can be cleaved in the cytoplasm were yet to be determined. 

Since the paper is very much oriented towards a future siRNA therapy, I think that this point needs to be clarified to the reader before publication, and that it will also help the reader focus on the new conjugation technique rather then the biological part. 

I would also remove or change the keyword Oligonucleotide therapeutics.  

Author Response

To address the reviewer's concern, we have added the following sentence to the end of the discussion:

Future studies are focused on: 1) whether or not the linker between the mAb and siRNA needs to be cleavable in the endosome prior to endosomal escape, which is the case for most ADCs[10], and 2) endosomal escape of the siRNA into the cytoplasm, which is the rate-limiting delivery step for achieving RNAi activity[1].